# Subpopulations of sensorless bacteria drive fitness in fluctuating environments

Thomas Julou[1,2]*, Ludovit Zweifel[1], Diana Blank[1,2], Athos Fiori[1,2], Erik van Nimwegen[1,2]*

**1** Biozentrum, University of Basel, Basel, Switzerland, **2** Swiss Institute of Bioinformatics, Basel, Switzerland

\* thomas.julou@unibas.ch (TJ); erik.vannimwegen@unibas.ch (EvN)

## Abstract

Populations of bacteria often undergo a lag in growth when switching conditions. Because growth lags can be large compared to typical doubling times, variations in growth lag are an important but often overlooked component of bacterial fitness in fluctuating environments. We here explore how growth lag variation is determined for the archetypical switch from glucose to lactose as a carbon source in *Escherichia coli*. First, we show that single-cell lags are bimodally distributed and controlled by a single-molecule trigger. That is, gene expression noise causes the population before the switch to divide into subpopulations with zero and nonzero *lac* operon expression. While "sensorless" cells with zero preexisting *lac* expression at the switch have long lags because they are unable to sense the lactose signal, any nonzero *lac* operon expression suffices to ensure a short lag. Second, we show that the growth lag at the population level depends crucially on the fraction of sensorless cells and that this fraction in turn depends sensitively on the growth condition before the switch. Consequently, even small changes in basal expression can significantly affect the fraction of sensorless cells, thereby population lags and fitness under switching conditions, and may thus be subject to significant natural selection. Indeed, we show that condition-dependent population lags vary across wild *E. coli* isolates. Since many sensory genes are naturally low expressed in conditions where their inducer is not present, bimodal responses due to subpopulations of sensorless cells may be a general mechanism inducing phenotypic heterogeneity and controlling population lags in switching environments. This mechanism also illustrates how gene expression noise can turn even a simple sensory gene circuit into a bet hedging module and underlines the profound role of gene expression noise in regulatory responses.

## Introduction

Most unicellular organisms live in variable environments that require them to adapt to, among other things, changes in available nutrients. Already in the early 1940s, Monod observed that during diauxic growth, when bacteria switch from consuming one sugar to another, there is typically a lag period in growth [1]. Reasoning that this lag is somehow associated with the need for cells to "adapt" to the alternative sugar, subsequent investigations into the origin of this lag eventually led to the discovery of gene regulation [2]. Monod also observed that the

**Data Availability Statement:** Raw microscopy datasets as well as preprocessed images obtained from mother machine experiments were deposited to the Image Data Resource under accession number idr0091 (https://doi.org/10.17867/

10000149). Data output from image analysis are available at https://doi.org/10.5281/zenodo. 3894719. Analysis scripts, as well as growth curves data and FLIM data, are available at https:// github.com/julou/MoM_lacInduction/releases/tag/ plos; the corresponding R working environment is provided as an attachment to this release (https:// git.io/JTS5A).

**Funding:** This work was funded by grant 31003A_159673 of Swiss National Science Foundation to EvN. The funders had no role in study design, data collection and analysis, decision to publish, or preparation of the manuscript.

**Competing interests:** The authors have declared that no competing interests exist.

**Abbreviations:** CPM, counts per molecule; DIMM, Dual Input Mother Machine; FCS, fluorescence correlation spectroscopy; FLCS, fluorescence lifetime correlation spectroscopy; FLIM, fluorescence lifetime imaging; GFP, green fluorescent protein; IPTG, isopropyl β-d-1-thiogalactopyranoside; PDMS, polydimethylsiloxane; OD, optical density; SPAD, single-photon avalanche diode; TCSPC, time-resolved, time-correlated single photon counting; TF, transcription factor; TMG, methyl-1-thio-β-D-galactopyranoside.

length of the lag can vary strongly for different sugars, suggesting that the amount of time cells need to adapt depends on the type of the switch, but at the time, it was not clear why certain switches would require more adaptation time than others.

It is currently widely believed that lag times depend on which of two types of general strategies cells employ to adapt to a particular environmental change. If, like in the *lac* operon studied by Monod, cells have dedicated machinery for sensing the nutrient and up-regulating the appropriate target genes in response, lags are generally expected to be short. Alternatively, in the absence of sensory machinery, cells can employ a bet hedging strategy where, through stochastic gene expression, small subsets of the population of cells are already preadapted to other environments [3]. In this case, long lags can result from the time required for the rare subset of preadapted cells to expand.

However, as it has become well appreciated that essentially all genes are subject to substantial stochastic fluctuations in expression, there may well be no clean dichotomy between sensing and stochastic bet hedging strategies [4]. That is, even when cells possess dedicated sensing and regulatory machinery for responding to a switch in nutrients, stochastic gene expression will cause phenotypes to vary across cells, and this may well lead to significant variations in lags across single cells.

The archetypical example of a nutrient switch, which led to the discovery of gene regulation, involves exposing *Escherichia coli* bacteria that were growing on glucose to lactose. In response, the three genes of the *lac* operon will be expressed, including the lactose transporter LacY and the galactosidase LacZ, which hydrolyzes lactose and isomerizes it into allolactose (Fig 1A). The expression of the *lac* operon is controlled by a dedicated transcription factor, LacI, which is inhibited by allolactose. In the absence of lactose, the repressor LacI keeps the operon at low expression. Mechanistically, although thermal fluctuations will cause LacI to occasionally unbind from the promoter at stochastic intervals, the high concentration of active LacI will cause it to rebind quickly, so that only very rarely RNA polymerase initiates transcription of the *lac* operon before LacI rebinds. However, in the presence of lactose, a positive feedback loop between import of lactose by lacY, production of allolactose by LacZ, and inhibition of LacI by allolactose, can cause the operon to switch to a high induced state [2]. Previous work has suggested that the rate-limiting step for switching of the promoter to the induced state is the stochastic waiting time for a sufficiently long period of the *lac* promoter remaining unbound by LacI, such that a sufficiently larger burst of *lac* transcription occurs and initiates the positive feedback loop [5].

We recently reported a preliminary investigation into the distribution of single-cell lags in the induction dynamics of the *lac* operon and observed that lags are bimodally distributed [6]. Here we characterize the molecular mechanisms underlying this bimodality, the determinants of the fractions of the population in each of the modes, and the resulting impact on population lag and fitness when bacteria are switching between nutrients.

Using a combination of experiments that perturb the state of cells before the first induction and detailed analysis of transcriptional memory in lineages of single cells after induction, as well as independent single-molecule fluorescence experiments, we show that cells with short and long lags correspond to cells with either nonzero or zero preexisting *lac* expression at the time of the switch. That is, we find that the *lac* operon constitutes a single-molecule trigger where any nonzero expression of LacY/Z proteins suffices to ensure a quick response. In contrast, long lags occur because cells with zero *lac* expression cannot sense lactose and have to wait for a spontaneous stochastic burst of *lac* operon expression before they induce. We show that the population lag is determined by the fraction of "sensorless" cells with zero *lac* expression and that this fraction can sensitively depend on the precise growth conditions before the switch. Using simulations to infer population growth lags from single-cell lags, we show that subtle changes in the fraction of sensorless cells can have substantial consequences for

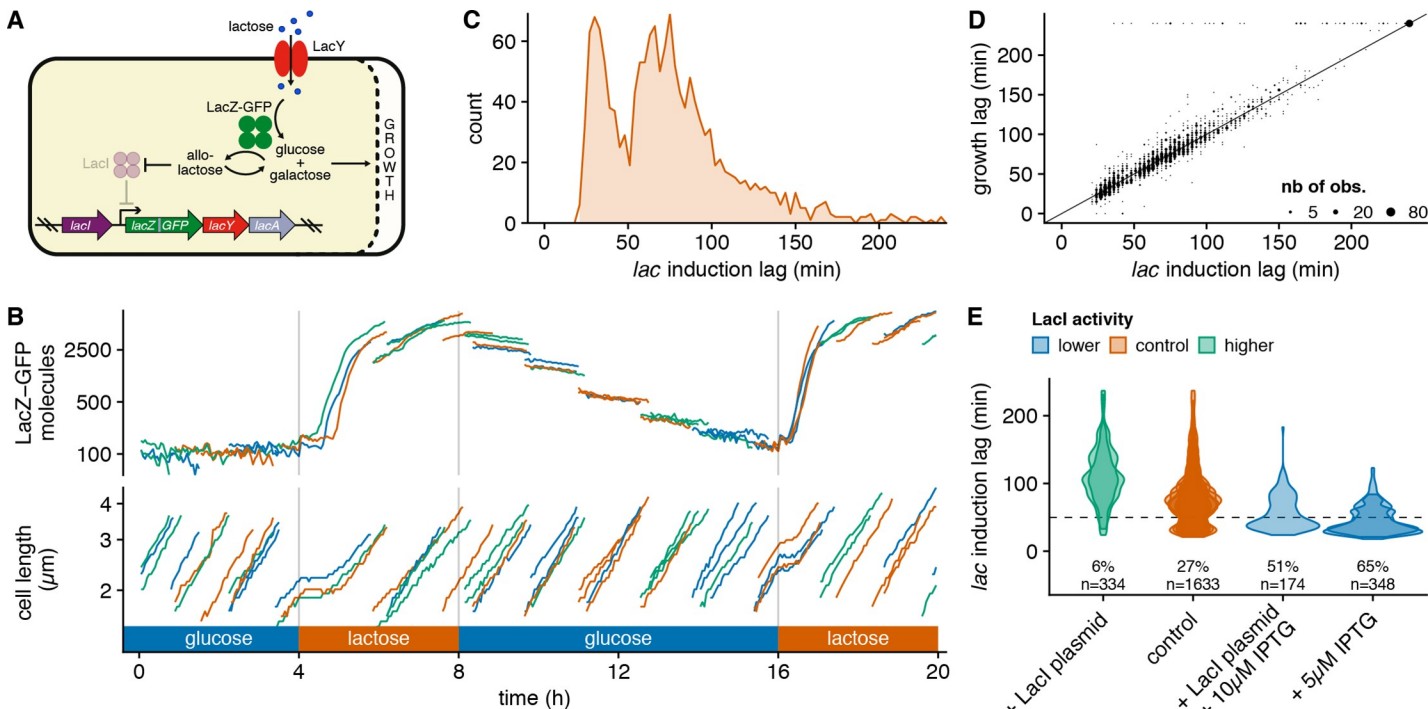

**Fig 1. *lac* operon induction lags are bimodally distributed and controlled by LacI activity before the switch (S1 Data, https://git.io/JTS5A).** (A) Summary of the *lac* gene circuit. We use a LacZ-GFPmut2 translational fusion integrated at the native genetic locus to monitor *lac* expression in single cells. (B) Time courses of cell length and number of LacZ-GFP molecules (on log scales; calibration as described in [6]) for a subset of cells from a typical experiment where cells are grown in a DIMM (S1 Fig) and exposed to 2 consecutive 4-h lactose episodes interspersed by a glucose period (which is 8 h in this experiment); for clarity, only the 4 cells near the closed end in one representative growth channel are shown using random colors to distinguish cells, and LacZ-GFP levels are offset by 100 molecules. (C) Histogram of single-cell induction lags for the *lac* operon at the first lactose exposure ("short" lags under 50 min and "long" lags above 50 min); *lac* induction lags were defined as the delay after the switch until cells increase their LacZ-GFP by 200 molecules and were estimated from time series of LacZ-GFP expression (shown in B) for 1633 cells in 9 independent replicates (S2 and S3 Figs). (D) A scatter of growth lags versus *lac* induction lags in individual cells shows that these two lags are tightly correlated (the solid line is a guide for the eye with *y* = *x*). (E) Violin plots of the distribution of single-cell induction lags for populations of cells whose LacI activity during growth in glucose was perturbed either by overexpression from a plasmid or by titration with sub-induction concentrations of IPTG. The fraction of cells with short lags is reported for each treatment; each contour corresponds to an independent replicate (with ≈150 cells), and the dotted line corresponds to 50 min separating long and short lags. DIMM, dual input Mother Machine; GFP, green fluorescent protein.

population lag and fitness. Diauxie experiments with both laboratory strains and a diverse set of wild *E. coli* isolates validate our predictions on population lags and show that wild *E. coli* strains also exhibit highly context-dependent lags in *lac* operon induction.

In summary, for the *lac* operon, a subpopulation of sensorless cells causes a bimodal response to lactose, and the population growth lag is largely determined by the fraction of sensorless cells at the time of the switch, which in turn sensitively depends on the growth conditions before the switch. This raises the question as to what extent the occurrence of sensorless cells is a general mechanism for generating bimodal responses and determining population lag to an inducing signal. Using existing proteomic data, we show that a large fraction of sensory systems in *E. coli* are expressed at such low levels in many conditions that substantial fractions of sensorless cells are expected to generically occur for many sensory systems. Thus, population lags may well be controlled by fractions of sensorless cells for many environmental switches.

## Results

In order to study the stochastic induction dynamics of the *lac* operon, we used a recently developed integrated setup that combines microfluidics, quantitative time-lapse microscopy, and

automated image analysis [6] (S1 Fig). We monitored growth and gene expression of single
*E. coli* cells expressing a translational LacZ-GFP fusion from their native *lac* operon locus (Fig
1A) while growing in Dual Input Mother Machine (DIMM) chips, where the growth medium
alternated between glucose and lactose, starting from a glucose phase (Fig 1B). As we already
reported in our preliminary study based on a much smaller sample [6], LacZ-GFP levels are
below autofluorescence levels in all cells during the initial glucose phase, and upon the switch
to lactose, all cells immediately enter growth arrest for a stochastic period of time, after which
growth recommences approximately at the same time as LacZ-GFP expression is first detected
(Fig 1B). Notably, although we do not observe bistability in the sense that almost all cells even-
tually induce their *lac* operon, the distribution of lag times until induction is clearly bimodal
(Figs 1C and S3); 30% of cells induce their *lac* operon in less than 50 min, while the other 70%
take between 1 and 3 h. The bimodality of single-cell lags is not dependent on the abrupt
nature of the switch, because an almost identical bimodal distribution was observed for experi-
ments in which cells are exposed to a gradual transition from glucose to lactose over 40 min
(S4 Fig). Another striking feature of this regulatory response is that, for virtually all cells, the
resumption of growth occurs almost exactly when *lac* expression becomes detectable (Fig 1B).
In order to rigorously quantify the correlation between *lac* induction and growth lags, we
devised a Bayesian inference procedure to estimate the growth lag of single cells from their
(noisy) elongation curves (see Estimation of growth lags). We observe that growth lags and
induction lags are very tightly correlated (Pearson correlation coefficient $r^2$ = 0.91, Figs 1D
and S3), with the exception of rare cells ($\approx$3%), which did not restart growing despite having
induced their *lac* operon. Consequently, the distribution of induction lags shapes the popula-
tion growth response when cells are exposed to lactose.

The fact that the growth lags closely track the *lac* induction lags of individual cells, together
with the fact that we did not observe any correlation between lags and basic physiological
parameters such as the growth rate of the cells or their cell cycle stage at the switch (S5 Fig),
strongly suggests that the *lac* induction dynamics is the main determinant of growth after the
switch. These observations led us to hypothesize that the lag time of a cell may be controlled by
its preexisting *lac* expression before the switch. To test whether the fractions of long and short
lags can be altered using a perturbation that only affects *lac* expression before the switch, we
added a minute amount of the artificial inducer IPTG (5 μM) to the glucose media before the
switch. Although this amount of IPTG is far too small to cause induction of the *lac* operon (S6
Fig), it should inhibit some of the LacI proteins, thereby decreasing the repression and increas-
ing preexisting *lac* expression. Indeed, we observe that this amount of IPTG is sufficient to
increase the fraction of short lags by 2.4-fold (Fig 1E). In contrast, when the LacI repressor is
overexpressed approximately 5-fold from a low copy number plasmid, leading to a stronger
repression of the *lac* operon, virtually all naive cells have long lags (Fig 1E). Moreover, LacI
overexpression can be compensated by IPTG supplementation, indicating that these perturba-
tions are indeed counteracting each other (Fig 1E). Since LacI only targets the *lac* operon [7], it
seems highly unlikely that these perturbations in LacI activity alter anything in the state of the
cells besides the strength of *lac* operon repression. These results thus strongly support that sin-
gle-cell lag times are controlled by preexisting *lac* operon expression. The fact that we observe
a bimodal distribution of lag times suggests that there exists a threshold of preexisting *lac*
expression that controls whether a cell has a short or long induction lag. In this interpretation,
this threshold expression level would correspond to the critical level of preexisting *lac* expres-
sion required for the positive feedback of the *lac* system to quickly drive cells to the induced
state, whereas cells with preexisting *lac* expression below this threshold would have to wait
until a stochastic burst of *lac* expression would drive the system over this threshold.

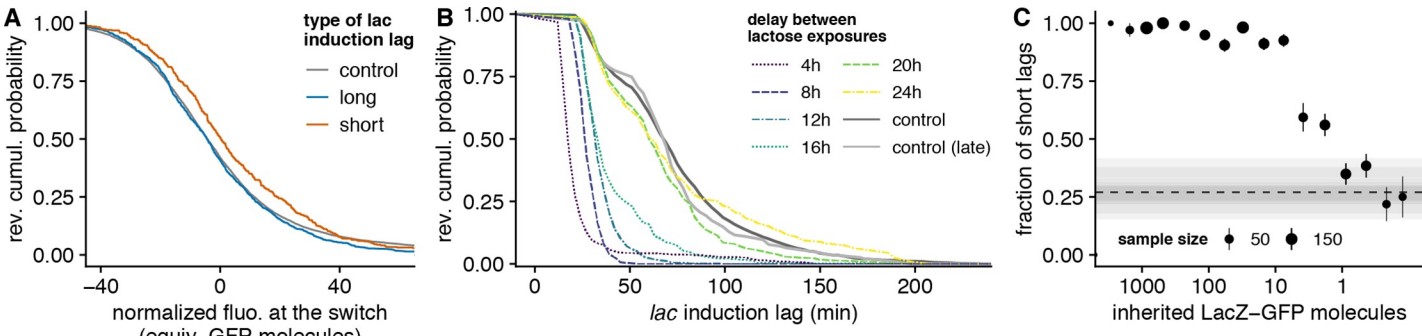

**Fig 2. Transcriptional memory analysis indicates that the critical *lac* expression threshold for inducing the *lac* operon is on the order of a single molecule (S1 Data, https://git.io/JTS5A).** (A) Distributions of initial fluorescence in cells with short (orange) and long (blue) *lac* induction lags; the gray line is a control showing the distribution of autofluorescence in a wild-type strain without GFP (fluorescence is centered per experiment, see Estimating LacZ-GFP levels before the switch). Note that the standard deviation of autofluorescence intensities corresponds to approximately 20 LacZ-GFP molecules, while the shift in fluorescence between cells with short and long lags corresponds to approximately 5 LacZ-GFP molecules only. (B) Reverse cumulative distributions of *lac* induction lags for cells that first grew in lactose and fully induced their *lac* operon and then grew in glucose for different lengths of time (shown in colors; 1 or 2 replicates per condition); the distribution for naive cells is shown as a reference (dark gray, corresponding to the histogram of Fig 1C) along with a control where cells are first exposed to lactose after 26 h in the microfluidic chip (light gray). (C) Fraction of short lags (<50 min) for cells preexposed to lactose as a function of the estimated number of inherited LacZ-GFP molecules. The dashed horizontal line shows the overall fraction of short lags in naive cells as a reference and semitransparent gray rectangles show the mean ± SE for each replicate. GFP, green fluorescent protein; SE, standard error.

We next set out to quantify this critical *lac* expression threshold value. Unfortunately, in our system, the fluorescence of preexisting LacZ-GFP molecules is small compared to autofluorescence levels, which fluctuate significantly from cell to cell (±20 GFP molecules; Fig 2A). In addition, small uncontrolled day-to-day variations in illumination contribute additional uncertainty on LacZ-GFP measurements. Altogether, this masks any correlation between preexisting LacZ-GFP levels and induction lags of individual cells (S5 Fig). However, by partitioning cells at the switch into those that exhibited either long lags or short lags, we find that cells with short lags have on average ≈5 LacZ-GFP molecules more than cells with long lags (*p*-value = 5.6e−06 using a 1-sided *t*-test; Fig 2A), indicating that the presence of only a few molecules may be enough to ensure short lags.

In addition, we were able to perform an independent and more precise quantification of the preexisting *lac* expression threshold using systematic dilution experiments. During the period of growth in glucose that intercedes the two lactose periods, the *lac* operon is fully repressed, and because LacZ-GFP is sufficiently stable that protein decay can be neglected [8], LacZ-GFP levels divide in half at each cell division (Fig 1B). For any cell at the second switch to lactose, we can thus estimate its LacZ-GFP level from the LacZ-GFP level of its ancestor at the end of the first lactose phase and the number of cell divisions in the intervening period. By modulating the length of the interceding glucose phase, we can thus systematically vary the distribution of remaining *lac* expression at the second switch to lactose and measure the fractions of long and short lags as a function of the level of remaining *lac* expression. We observe that even after surprisingly long periods of up to 12 h between consecutive exposure to lactose, virtually all lags are short, and that it takes ca. 20 h (corresponding to circa 16 divisions) to relax back to the distribution of induction lags observed in naive cells (Fig 2B).

Moreover, there is a very strong correlation between the observed lag times and the estimated number of remaining LacZ-GFP molecules (S7 Fig). Virtually all cells that are estimated to have at least 10 remaining LacZ-GFP molecules exhibit short lags, and only cells with 1 or fewer remaining LacZ-GFP molecules exhibit the same fraction of long lags as naive cells (Fig 2C). Together, these dilution experiments and direct fluorescence measurements (Fig 2A) indicate that the critical threshold of preexisting *lac* expression is on the order of 1 molecule.

Both the small shift in the distribution of fluorescence levels and the dilution experiments suggest a critical LacZ-GFP level on the order of a single molecule. To confirm this low threshold using a third independent method, we employed fluorescence lifetime correlation spectroscopy (FLCS) in order to achieve single-molecule sensitivity and at the same time to distinguish weak GFP fluorescence from autofluorescence in single cells (see Fluorescence lifetime experiments). We first characterized the lifetime distribution of GFP by measuring LacZ-GFPmut2 bacteria induced with 200 μM IPTG (hence making the relative contribution of autofluorescence negligible) and found a single-exponential decay with characteristic lifetime $\tau \approx 2.7$ ns. Similarly, to characterize the lifetime distribution of autofluorescence, we used wild-type bacteria not carrying GFP and found a double-exponential decay ($\tau_1 \approx 0.75$ ns, $\tau_2 \approx$ 3.8 ns; Fig 3A). Using these lifetime distributions, we were able to separate the individual fluorescence contributions in point FLCS measurements of uninduced LacZ-GFPmut2 bacteria and determine which cells had a nonzero presence of LacZ-GFP (see Fluorescence lifetime experiments; S9 Fig).

First, we measured snapshots of 150 uninduced bacteria cells from 5 independent liquid cultures, whose LacZ-GFP content is expected to be the same as in mother machine experiments before the first switch from glucose to lactose. Indeed, the proportion of cells where LacZ-GFP is detectable by point FLCS (28 ± 4%) is similar to the proportion of cells with short lags in microfluidic experiments determined by imaging (30 ± 1%, Fig 3B). Notably, this is consistent with previous single-molecule experiments, which observed that ≈65% of cells have no LacZ [9] and ≈50% of cells have no LacY [10]. Second, we performed three independent microfluidic experiments where we acquired FLCS data for 1 to 2 cells per growth channel at the onset of the switch from glucose to lactose. Additionally, we monitored the bacterial growth and the LacZ-GFP expression for 270 min by fluorescence lifetime imaging (FLIM) in order to resolve the induction lag in these cells. In line with our expectations, we found that only 1 out of 37 cells with a long lag had been measured to contain nonzero LacZ-GFP before the switch (Fig 3C). Conversely, we detected low but nonzero molecular levels of LacZ-GFP (fitted number of fluorescent particles <20) in the majority of cells with short lags (13 out of 22). The fact that not all cells with short lags were detected to contain nonzero LacZ-GFP likely reflects that our method lacks sufficient sensitivity to reliably detect a single LacZ-GFP tetramer and that our data analysis is conservative in calling nonzero LacZ-GFP levels. FLCS measurements in snapshot and time course experiments therefore support that the critical

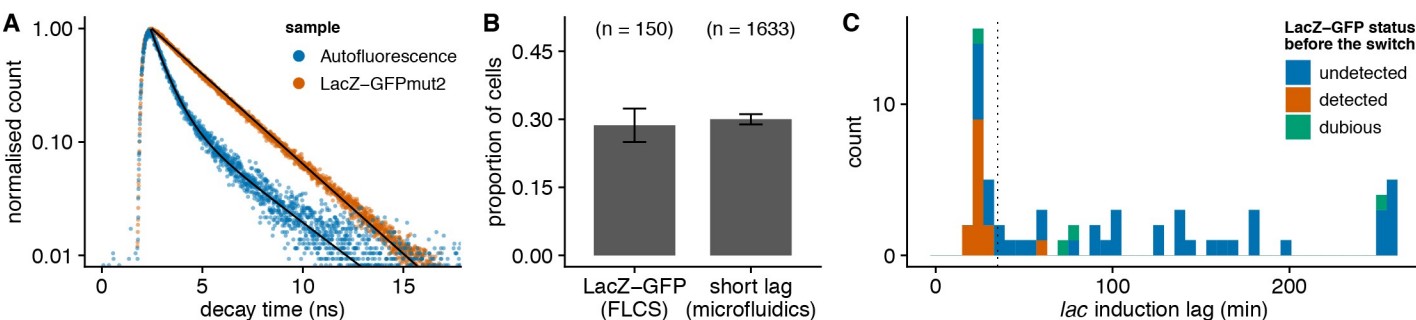

**Fig 3. Fluorescence lifetime analysis supports that the critical *lac* expression threshold for inducing the *lac* operon is on the order of a single molecule (data from https://git.io/JTSdM).** (A) Distribution of decay times measured on induced bacteria (LacZ-GFPmut2) and on bacteria without GFP (autofluorescence in GFP channel). The peak at 2.5 ns corresponds to the laser pulse; hence, the fluorescence lifetime is the delay after this peak. (B) Fraction of cells where LacZ-GFP is detected by FLCS in a liquid culture of uninduced cells (*n* = 150). The fraction of cells with short lags in microfluidic experiment (Fig 1C) is shown for comparison. (C) Distribution of *lac* induction lags during experiments where the presence of LacZ-GFP before the switch (shown in color) is assessed by FLCS (66 cells from 3 independent replicates). The dotted vertical line indicates the threshold between short and long lags. FLCS, fluorescence lifetime correlation spectroscopy; GFP, green fluorescent protein.

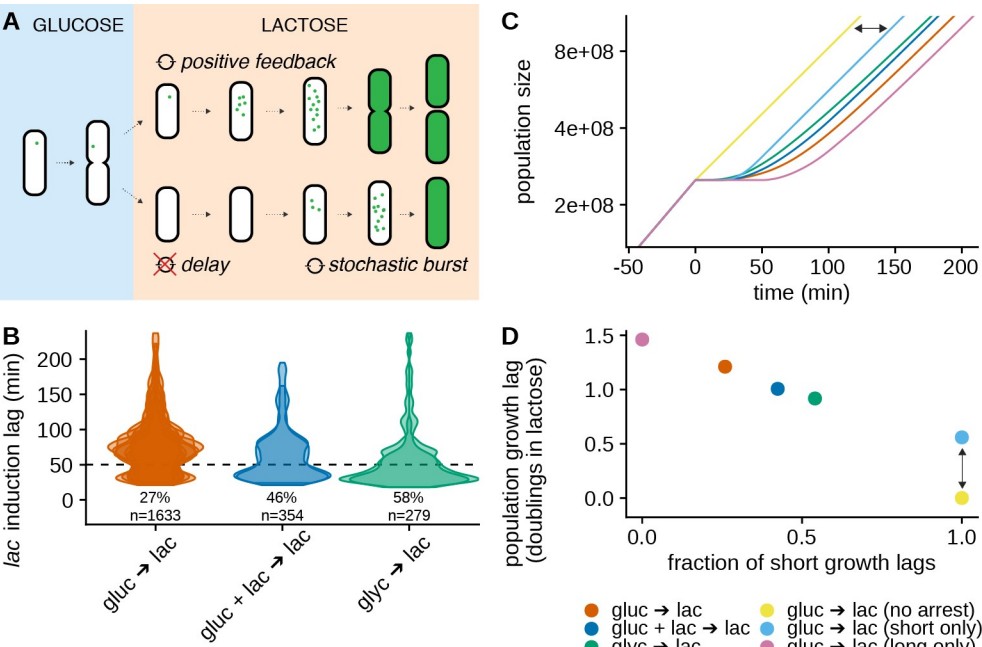

**Fig 4. The response of single cells to lactose is context-dependent and shapes the population fitness after the switch** (**S1 Data, https://git.io/JTS5A**). (A) Cartoon depicting how the *lac* induction of a given cell is determined by the stochastic expression in the repressed state and hence depends on the conditions before the switch. Green dots depict Lac proteins. (B) Violin plots of the single-cell lag distributions for switches to lactose from growth conditions in which the *lac* operon is slightly less repressed than in pure glucose, i.e., 0.4% glycerol (blue) and 0.2% glucose + 0.2% lactose (green). Each contour corresponds to a separate replicate with ≈150 cells; the fraction of short lags (<50 min, indicated by the dotted line) is indicated below the plots; the lag distribution measured in the glucose to lactose switch (as Fig 1C) is shown in red for comparison. (C) Simulated population growth curves for different distributions of single-cell lags, as observed for different switches (different colors; with semi-log scale). The simulations assume deterministic exponential growth of each cell before and after the switch, and complete growth arrest during the lag. The pink and light blue curves show the population growth that would be obtained with only short and long lags, respectively. (D) Population growth lags as inferred from the simulated population growth shown in panel C as a function of the fraction of short lags. The lag is measured as a delay compared to a population with no arrest (i.e., where all cells grow immediately at their maximum growth rate on lactose; yellow) and reported in number of doublings in lactose (i.e., the number of divisions lost due to growth arrest). For example, the vertical arrow indicates the lag for the light blue condition and corresponds to the delay shown by the horizontal arrow in panel C. Note that "short only" (light blue) and "long only" (pink) correspond to hypothetical populations where either all cells have long lags, or all cells have short lags.

threshold of preexisting *lac* expression is on the order of 1 molecule. While single-cell fluorescence correlation spectroscopy (FCS) studies in bacteria are not uncommon, this is to our knowledge the first report where gene expression noise is studied by assessing the presence/absence of freely diffusing fluorescent proteins in live bacteria by FLCS.

Taking these results together, the following picture of a single-molecule trigger emerges (Fig 4A). Since it is well known that the LacY permease is required for import of lactose into the cell [11] and LacZ is required for production of allolactose, cells that lack any molecules of either LacY or LacZ cannot sense the lactose in the environment and cannot induce. However, as soon as one or just of few molecules of LacZ and LacY are present, cells can respond fast. Note that, although we have not directly quantified the threshold number of LacY molecules necessary to ensure a fast response, the fact that essentially all cells with at least 10 molecules of LacZ respond fast indicates that LacY levels must have been over the threshold in all cells with at least 10 molecules of LacZ. Given that LacY and LacZ are co-expressed from the same operon, this strongly suggests that the threshold number of LacY molecules must be small as well.

Cells that lack either LacZ or LacY have to wait for a stochastic burst of *lac* operon expression in order to induce which, as suggested previously [5], depends on the stochastic waiting time for LacI to unbind from the promoter for sufficiently long to allow transcription to occur. However, our results suggest that threshold levels of LacZ and LacY are so small that even a single transcript of the *lac* operon may suffice, at least in the conditions tested here.

To investigate to what extent the low threshold in the number of LacZ molecules depends on the external lactose concentration, we performed additional experiments with both 200-fold and 20-fold lower lactose concentrations. At 200-fold lower lactose concentration, no induction was observed, and all cells remained growth arrested for more than 12 h, suggesting that cells cannot induce the *lac* operon at such low external lactose concentrations. At 20-fold lower lactose concentration, we again observed a bimodal distribution of lag times, and although induction times were slightly longer, probably due to the slower import of lactose, the threshold number of LacZ-GFP molecules did not increase appreciably from what was observed at higher lactose concentration, i.e., 1 to 10 molecules (S8 Fig; see Estimating the LacZ-GFP threshold for induction at low lactose concentration). These results support that the native *lac* operon responds to lactose as a single-molecule trigger in a wide range of conditions.

One implication of the single-molecule trigger mechanism is that the fractions of short and long lags depend sensitively on the strength of the *lac* operon repression before the switch, i.e., on the fraction of cells that have precisely zero expression. This suggests that even subtle changes in the strength of *lac* operon repression due to changes in growth conditions before the switch may substantially affect the distribution of single-cell lags. To investigate the context-dependence of the distribution of single-cell lags, we used different nutrients before the switch that we reasoned would alter the strength of repression. In particular, besides being repressed by LacI, the *lac* operon is also positively regulated by the transcription factor CRP [12], and the activity of CRP can be increased by, for example, replacing glucose with glycerol as a carbon source [13]. Since the strength of the *lac* repression should thus be weakened when growing in glycerol, the fraction of short lags should increase, and this is indeed what we observe, i.e., the fraction of naive cells inducing fast increases 2.1-fold to 58% (Fig 4B). In addition, we also measured single-cell lags when growing cells on a mixture of glucose and lactose before the switch, i.e., as in diauxie experiments. Although *lac* expression remains low on such a mixture [14], whenever the *lac* expression is nonzero, the presence of lactose will cause some of it to be transported into the cells and metabolized into allolactose, inhibiting some fraction of the LacI repressors. Thus, compared to growth on glucose, we expect the repression of the *lac* operon to be slightly weakened on a mixture of glucose and lactose. Note that the direct inhibition of LacY permeases due to the ongoing glucose import ("inducer exclusion") is not complete and should thus not prevent this effect from occurring [12]. Indeed, we find that under these conditions, the fraction of short lags is increased 1.8-fold to 48% (Fig 4B). These results confirm that, because induction depends on a single-molecule trigger, the fraction of cells with short lags is highly sensitive to the precise growth conditions before the switch.

The sensitive dependence of the single-cell lags to environmental conditions will translate into differences in population lags and thus potentially in fitness. However, the magnitude of these potential effects on fitness is not a priori clear.

In order to infer population growth lags from distributions of single-cell growth lags, we simulated population growth curves during a switch to lactose using the observed distribution of single-cell lags in a given condition (Fig 4C). By comparing these to the growth curve of a population where all cells grow immediately at their maximum growth rate on lactose (yellow line in Fig 4C), we can calculate a population lag for each of the conditions described above (Fig 4D). In addition, we calculate population lags for hypothetical populations where cells

would have only short lags or only long lags. Remarkably, while a population where all cells switch fast will have a lag on the order of half a division time (due to the time necessary to initiate the positive feedback), a population of naive cells will be delayed by approximately 1.2 divisions compared to a population without lag, and the population lag can be as long as 1.5 division times if no cells have Lac proteins at the switch. Thus, we find that, by affecting the fraction of sensorless cells with zero *lac* expression, even subtle changes in growth conditions before the switch can have a strong impact on population lag.

Our measurements of the single-cell lags suggest that population lags, and thus fitness in response to a switch of the available nutrients to lactose, can sensitively depend on the growth conditions before the switch. To test this in real populations, and to investigate to what extent this behavior extends beyond lab strains, we performed classic diauxie experiments with both lab strains and a diverse set of natural *E. coli* isolates [15,16]. As is well known, when *E. coli* is grown on a combination of glucose and lactose, cells first consume the glucose and naturally switch to consuming lactose when the glucose runs out. To test the context-dependence of population lags, we performed diauxie experiments with two different lactose concentrations.

Based on our previous observations with experiments in microfluidics (Fig 4B), we reasoned that higher external concentrations of lactose would lead to more lactose being imported into the cells during growth on glucose, thereby weakening the *lac* repression more. Thus, we expect shorter population lags in the higher lactose concentration. To calculate growth lags, we compared the observed diauxic growth curves with population growth in the same conditions supplemented with large amounts of IPTG (200 μM), such that cells express their *lac* operon during the whole experiment and can readily metabolize lactose when they exhaust glucose (Fig 5A). Across all strains, we find that population lags are indeed shorter for higher lactose concentrations, even though the absolute value of the lags and the size of the difference between the two lactose concentrations vary substantially across strains (Figs 5B and S10).

Although, based on our single-cell experiments, we interpret these changes in population lags to result from changes in the fractions of cells with nonzero *lac* expression at the time that the glucose is exhausted, a higher lactose concentration may also reduce lags by increasing the rate of lactose import in cells with nonzero *lac* operon expression, i.e., shortening the time of

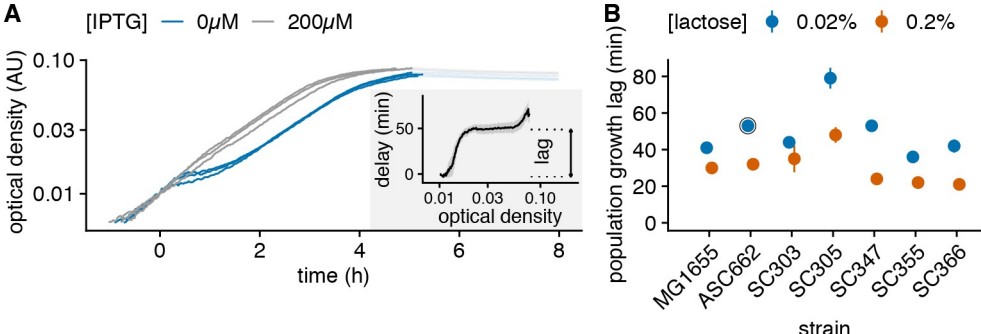

**Fig 5. Single-molecule triggering is observed across a diverse set of *E. coli* strains (data from https://git.io/JTSd5).** (A) Estimation of the population growth lag from diauxie experiments. Population growth curves of strain ASC662 during diauxie experiments with 0.02% lactose and 0.005% glucose (blue) and controls where the *lac* operon is expressed throughout with supplemented ITPG (gray). Each line corresponds to 1 replicate. The delay between the blue growth curve and gray control to reach a given optical density features a long plateau, which corresponds to the population lag (inset). (B) Population lag in diauxie experiments with medium (0.02%, blue) and high (0.2%, orange) lactose concentrations for both lab strains and a diverse set of natural *E. coli* isolates. Points and error bars correspond to mean and standard error over at least 3 replicates with 0.005% glucose; the circled dot correspond to the growth curves shown in panel A. Note that the lag is reduced in all strains when more lactose is provided.

the induction process itself. To disentangle these effects, we performed experiments in which cells were grown on a mixture of glucose and lactose (at a given concentration) and were then transferred to media with only lactose at varying concentrations. Despite the less precise measurement of the population lag with this protocol than in diauxie experiments, we observe that the growth lag after the switch is impacted both by the lactose concentrations before and after the switch, and that these 2 parameters have effects of similar magnitude on the population lag (S11 Fig). Altogether, these experiments in bulk indicate that across a wide range of *E. coli* strains, the *lac* operon basal expression is so low that it functions as a single-molecule trigger.

Our experiments have shown that population lag times, and thus fitness under an environmental switch to lactose, are to a large extent determined by the frequency of sensorless cells with zero *lac* operon expression. This observation raises the question if other sensory systems in *E. coli* are also repressed to the point that substantial fractions of sensorless cells will be unable to respond when the signal appears. We are aware of only one other system that has been shown to exhibit very similar behavior and that is the regulatory response protecting *E. coli* against the toxic and mutagenic effects of DNA alkylation damage. In this system, a single-molecule trigger leads to fast or slow expression of Ada proteins, which transiently creates two subpopulations with different phenotypes, in this case, mutation rates [17]. In order to address how ubiquitous single-molecule triggers are in bacteria, we focused on two-component systems. Two-component systems all feature a kinase sensor located at the cell envelope whose activation by environmental signals lead to phosphorylation of a cognate response regulator, typically a transcription factor (TF). In many cases, the activation of the response regulator up-regulates the expression of the kinase, thereby implementing a positive feedback. Thus, two-component systems with very low basal expression of the kinase could exhibit single-molecule trigger dynamics. But even without single-molecule trigger behavior, sufficiently low basal expression of the kinase will cause sensorless cells to appear, and they will be unable to respond to the corresponding signal.

Using quantitative proteomics data obtained for *E. coli* in 28 conditions [18], we could query the expression levels for 9 kinase-TF pairs and for another 12 TFs (S12 Fig), out of 27 two-component systems annotated on ecocyc.org. In order to assess the fraction of sensorless cells (i.e., cells with zero protein) from these bulk measurements of the **average** expression levels, we conservatively assumed Poisson expression fluctuations across cells and focused on conditions where at least 5% of the population is sensorless; this condition is met when the average protein level is 3 copies or less (since, for a Poisson distribution of mean $\lambda$, $P(N = 0) = e^{-\lambda}$ and $e^{-3} \approx 0.05$). Remarkably for 4 of the 9 pairs, cells carry less than 3 copies of the kinase on average in many conditions (Fig 6). It is also remarkable that for

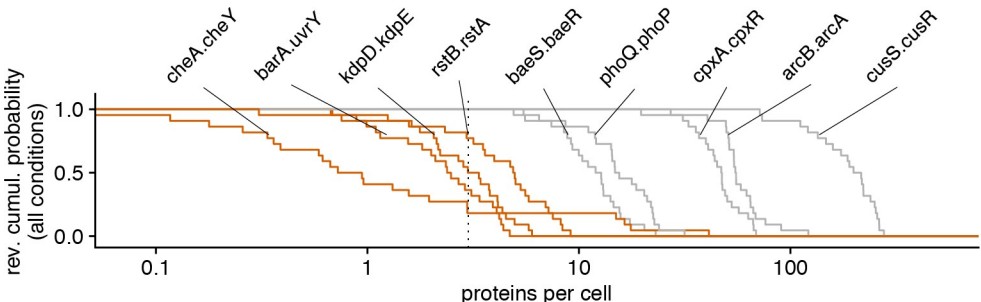

**Fig 6. Sensorless cells are expected to occur for many two-component systems in *E. coli*.** Distributions of average expression levels of sensor kinases across 28 conditions for 9 of the 28 annotated two-component systems in *E. coli*. Assuming Poisson fluctuations in expression, at least 5% of the cells are expected to be sensorless when the average kinase level is 3 proteins per cell or below (vertical dotted line). Note that 4 out of 9 two-component systems (highlighted in orange) feature sensorless cells in 6 conditions or more (data from [18]).

all 9 pairs, the kinase is expressed at a lower level than the TF (typically 200- to 1,000-fold less); since 6 out of the 12 additional TFs are expressed at 10 copies or less per cell (S12 Fig), we expect several of these two-component systems to have sensorless cells as well. This limited exploration of sensory gene circuits in *E. coli* illustrates that the occurrence of sensorless cells and single-molecule triggers may in fact be common in bacteria.

## Discussion

In this study of the *lac* operon induction by lactose in single cells, we uncovered that it is governed by a single-molecule trigger; while cells that have at least 1 LacZ protein (and also LacY proteins) expressed at the moment of the switch will respond readily, cells lacking Lac proteins are unable to sense lactose and have to wait for the next stochastic burst of *lac* expression, which can delay the response for as long as several divisions. This mechanism, where a positive feedback involved in the response is combined with a very low basal expression level of its sensor such that a fraction of cells is effectively sensorless, gives rise to transiently coexisting induced and uninduced subpopulations. Importantly, despite involving a positive feedback, this molecular mechanism underlying phenotypic variability does not require bistability of the gene circuit, as highlighted by the fact that all cells eventually switch on.

The *lac* operon is one of the most widely studied regulatory systems, which raises the question why this single-molecule trigger has not been observed previously. In most previous quantitative studies of the *lac* operon, artificial inducers such as IPTG and TMG were used in order to measure *lac* expression when cells are in a steady state of growth that is not perturbed by the induction [12,19]. In these conditions, true bistability has been reported, with a critical threshold on the order of hundreds of Lac proteins [10]. We note two previous studies of the *lac* induction dynamics that are most closely related to our current study where the inducer is lactose. Lambert and Kussell studied *lac* induction by lactose in small populations of *E. coli* using a microfluidic setup where bacteria are directly exposed to switching growth media [8]. However, in contrast to our study where growth and gene expression is followed at the single-cell level, growth is only tracked for the entire population. This study established the long-lived memory of *lac* operon induction, i.e., that a population of cells is able to respond rapidly as long as Lac proteins have not been diluted below a threshold. However, the setup was not calibrated to measure *lac* expression in number of proteins so that this threshold could not be quantified, and the bimodality of induction lags could not be uncovered because of the lack of single-cell measurements. In a second study, *lac* operon induction was studied in single cells by growing cells in microfluidic devices in which media were dispensed by diffusion through a gel, and growth lag upon a switch to lactose was measured in single cells [20]. Because no growth arrest was observed for the large majority of cells in this study, lags were defined as the delay until the growth rate has fully recovered, and no bimodality of this recovery time was observed. We hypothesize that, due to the different method of nutrient switching, cells may have been exposed to remaining low levels of glucose for sufficiently long to avoid growth-arrest in this study.

Thus, both the precise temporal control of the growth media in our DIMM setup and also the accurate quantitative tracking of growth and gene expression in lineages of single cells were essential for uncovering that a single-molecule trigger underlies the bimodal behavior of *lac* operon induction and that it constitutes the molecular basis of cellular memory after exposure to lactose. In addition, we showed that the population lag is determined by the fractions of slow- and fast-responding cells and that these fractions are highly sensitive to growth conditions before the switch.

Overall, we provide here the first example of how a gene circuit with positive feedback that implements a sense-and-response strategy can also effectively implement a bet hedging

strategy in response to environmental changes because of the occurrence of a fraction of sensorless cells.

The occurrence of a subpopulation of sensorless cells can have major consequences for fitness under switching environments, by causing substantial population lags and delaying growth by more than an entire division. Moreover, substantial population growth lags as a result of a subpopulation of sensorless cells may be a fairly common feature of bacterial sensory systems, as our survey of two-component systems shows that, for a large fraction of them, sensors are so low expressed that a sizable fraction of cells with zero expression is to be expected. This observation, in combination with the observation that the fraction of sensorless cells is highly sensitive to the strength of repression of the system, raises the question as to how natural selection has acted on the expression levels of sensory systems. Indeed, as our experiments with sub-inducing amounts of IPTG demonstrate, even very modest derepression of the system would dramatically lower the fraction of sensorless cells, and it seems that bacteria could easily evolve to do so.

However, we observed that substantial population lags occur in all strains of a diverse set of natural *E. coli* isolates. The fact that sensorless cells have apparently been maintained across a diverse set of wild isolates suggests that there must be a cost to even a very low level of expression of a sensory system in the absence of its cognate signal, raising the question what that cost might be. It has been argued that sensing may generally incur an energetic cost [11]. Nonetheless, it seems unlikely to us that very low levels of *lac* operon expression would incur a significant energetic cost. Alternatively, the cost may derive from toxicity (e.g., due to biochemical or transport activity of the sensor). Although this may be plausible for some sensory systems, e.g., the Ada protein involved in DNA repair [17], we are not aware of any candidate molecules transported by the LacY permease that would be toxic for the cell when only a few copies of LacY are expressed. A third possibility is that because of general constraints on resources, it is simply impossible for cells to express **all** of its many sensory systems at nonzero levels in parallel. Consequently, each individual cell would be forced to be sensorless for a subset of the sensory systems, and expressing one sensory system would go at the cost of not being able to express some others.

It is tempting to speculate that natural selection may have tuned the expression levels of the sensory systems to ensure that while the majority of cells may be sensorless for any given sensor, each sensor is present in a small subpopulation, thereby providing an effective bet hedging strategy for limiting population lags under a sudden appearance of the signal. Finally, we have seen that the fraction of sensorless cells is highly sensitive to the precise growth conditions. Speculating even further, this raises the possibility that natural selection might be able to tune the sensor's expression in a condition-dependent manner, so as to effectively anticipate the appearance of the signal in some conditions more than in others. For example, it has been observed that *mal* operons have higher basal expression on lactose than on other sugars, leading to a shorter population lag after a switch to lactose [21]. Similarly, in the case of the *lac* operon, because CRP activity increases whenever carbon flux is limiting [22], this will cause a slight derepression of not only the *lac* operons, but also all operons of alternative carbon sources regulated by CRP, thereby effectively anticipating appearance of any of these alternative carbon sources. Further studies will be required to establish to what extent such stochastic anticipatory strategies implemented by tuning of the fractions of sensorless cells are widespread among bacteria.

## Materials and methods

### Bacterial strains and media

The *E. coli* strain used in the microfluidics experiments is ASC662 (MG1655 lacZ-GFPmut2; obtained from Daan Kiviet) [23]. We confirmed by sequencing that this strain has no mutation

in the *lac* operon apart from the lacZ-GFP translational fusion and identified the GFP sequence as a GFPmut2 lacking the monomeric substitution A206K; fortunately, this does not affect its spectrum nor its maturation time [24].

Diauxie experiments were done with this strain ASC662 and with its parent strain MG1655 (CGSC #7740) as well as with 6 natural isolates of *E. coli* [15]: SC303, SC305, SC330, SC347, SC355, and SC366.

All experiments were done using M9 minimal media (Sigma-Aldrich) supplemented with 2 mM MgSO4, 0.1 mM CaCl2, and sugars as indicated (typically 0.2% for glucose or lactose and 0.4% for glycerol). TMG and IPTG were diluted from frozen stocks in water (at 0.1 M and 1 M, respectively).

All experiments were carried out at 37˚C. Note that the melAB operon is not expressed at this temperature [25], so that it should not interfere with the regulation of the *lac* operon.

In order to increase LacI activity in the repressed state (Fig 1E), plasmid p_lacI_SC101 was transformed into ASC662. It was obtained by restriction-ligation cloning of MG1655's native lacI promoter and gene (amplified by PCR between positions 366,428 and 367,601) into a pSC101 vector (pUA66; using XbaI and XhoI, hence removing the plasmid's promoter and GFP loci). In order to make cloning easier, we used a high copy number derivative of pUA66 where the GFP locus is replaced with pUC19 origin (produced by restriction-ligation cloning with BamHI and XbaI).

## Microfluidics experiments

**Microfluidic device fabrication.** The DIMM microfluidic design used in this study has been described elsewhere [6] and is freely available online [26]; in brief, it combines comb-like structures enabling to grow bacteria in steady-state conditions over long times with a special type of junction allowing fast and accurate mixing of two input media (S1 Fig). Several microfluidics masters were produced using soft lithography by micro-resist Gmbh; one master with regular growth channels of suitable size (0.7 μm width × 0.9 μm height) was used for all experiments.

For each experiment, a new chip was produced by pouring polydimethylsiloxane (PDMS) (Sylgard 184 with 1:9w/w ratio of curing agent) on the master and baking it for 4 h or more at 80˚C. After cutting the chip and punching inlets, the chip was bonded to a #1.5 glass coverslip as follows: the coverslip was sonicated 3 to 5 min in acetone and rinsed in isopropanol and then in water; the chip was cleaned from dust using MagicTape and rinsed in isopropanol and then in water; surfaces were activated with air plasma (40 s at 1500 μm of Hg) before being put in contact; the assembled chip was cooked 1 h or more at 80˚C.

Before running the experiment, the chip was primed and incubated 1 h at 37˚C using passivation buffer (2.5 mg/mL salmon sperm DNA, 7.5 mg/mL bovine serum albumin) for the mother machine channels and water for the overflow channels.

**Culture conditions and flow control.** Bacteria were streaked onto LB agar plates from frozen glycerol stocks thawed from −80˚C. Overnight precultures were grown from single colonies in M9 minimal medium supplemented with the same sugar that the cells were to experience in the initial condition of the experiment. The next day, cells were diluted 100-fold into fresh medium with the same sugar and harvested after 4 to 6 h.

The experimental apparatus was initialized, prewarmed, and equilibrated. Flow control was achieved using a syringe pump for early experiments and using a pressure controller later on. A total flow of ≈3 μL/min was used in all cases (corresponding to a pressure of ≈2,000 mbar on both inlets). Polystyrene beads (Polybead, Polysciences, 1 μm) were added to one of the two media to allow observing the flow ratio at the mixing junction.

The primed microfluidic chip was mounted, connected to media supply, and flushed with running media for 30 min or more to rinse passivation buffer. The grown cell culture was centrifuged at 4,000×g for 5 min, and the pellet resuspended in a few microliters of supernatant and injected into the device using a syringe. Since flow was running, the pressure had to be continuously adjusted to make sure the cells stopped flowing in the main channel and could enter the growth channels (typically 10 to 40 min).

After loading, bacteria were incubated during 2 h before starting image acquisition. Every 3 min phase contrast and fluorescence images were acquired for several positions in parallel (typically 6), including the dial-a-wave junction with short phase contrast exposure (10 ms) so as to monitor the flow ratio between the 2 inputs.

All conditions used in this project are summarized in S1 Table. The default protocol consists of 6 h in M9 + 0.2% glucose followed by 4 h in M9 + 0.2% lactose.

**Microscopy and image analysis.** An inverted Nikon Ti-E microscope, equipped with a motorized *xy* stage, and enclosed in a temperature incubator (TheCube, Life Imaging Systems) was used to perform all experiments. The sample was fixed on the stage using metal clamps, and focus was maintained using hardware autofocus (Perfect Focus System, Nikon). Images were recorded using a CFI Plan Apochromat Lambda DM ×100 objective (NA 1.45, WD 0.13 mm) and a CMOS camera (Hamamatsu Orca-Flash 4.0). The setup was controlled using μManager [27] and time-lapse movies were recorded with its multidimensional acquisition engine (customized using runnables). Phase contrast images were acquired using 100 ms exposure (pE-100, CoolLED, full power). Images of GFP fluorescence (ex 475/35 nm; em 525/50 nm; bs 495 nm) were acquired using 2 s exposure (SpectraX, Lumencor, Cyan channel at 17% with a ND4 neutral density filter).

Image analysis was performed using the software MoMA [6] as described in its documentation [28]. Raw image datasets (available at https://doi.org/10.17867/10000149) were transferred to a centralized storage and preprocessed in batch. For each experiment, 30 growth channels were picked randomly (after discarding channels with structural defects or no cells growing at the first switch of condition) and curated manually in MoMA (data available at https://doi.org/10.5281/zenodo.3894719). MoMA's default post-processing was used in order to refine the measurements of cell length and total fluorescence. Fluorescence arbitrary units were converted to the number of GFP molecules using the procedure and conversion factors described previously [6].

**Quality control and data filtering.** We established previously that cells slightly decrease their growth rate (approximately 10% to 15%) in response to our illumination conditions and reach a steady state of growth after 2 h [6]. Accordingly, the first 2 h were discarded from further analysis in all experiments.

Given the design of the DIMM microfluidic chip, only 1 strain and 1 condition can be used in each experiment. As a result, this study is based on more than 30 independent experiments performed over almost 3 years (S1 Table). In order to minimize the confounding effect that the initial state of cells might affect their response to environmental change, we discarded experiments showing atypical growth patterns (used as a proxy for physiological state) before the switch of condition occurred. For each full cell cycle observed before the first switch, we fit a linear model to log(length) as a function of time, and extract from this fit the growth rate, the Pearson correlation coefficient of the fit, and the predicted initial cell length. The distributions of correlation coefficients did not exhibit systematic changes across dates. However, the growth rate and initial cell length showed some variability across dates, and, as expected, there was a clear correlation between cell size and growth rate in each condition (S2 Fig). The figure shows that 3 experiments appear as obvious outliers with slow-growing cells, and these were discarded (20151218, 20161021, and 20180313); in addition, we discarded 2 more experiments

with slow-growing cells (20160526 and 20170108). Finally, we also examined the reproducibility of the *lac* induction lags distributions between replicates and closely related conditions and discarded 2 additional experiments because of their aberrant distributions of lags (20180123 and 20180615). No reason could be found for these last 2 experiments to be qualitatively different from our expectations (leaving the door open to an undocumented protocol mistake).

In addition, since our study focuses on the response of **growing** cells to changes of environmental conditions, we checked that the fraction of cells not growing before the switch in the analyzed growth channels is small in all experiments (<5).

**Estimation of *lac* induction lags.**   The time until the *lac* operon is induced in a given cell after a switch to lactose is estimated from the time series of LacZ-GFP level as follows: We compute the preinduction level computed as the mean value in the 9 min following the switch (for which we checked visually that no induction occurred) and measure the delay after the switch until the cell has increased its level by 200 molecules (which is a conservative threshold since the fluctuations of background level have a standard deviation of ≈20); hence, we need to observe a cell for at least 4 time points after the switch to be able to successfully detect an induction. In order to successfully estimate a lag for as many cells as possible, if a cell divided before inducing the total GFP of its two daughters was summed, and cell traces were considered until the induction threshold were passed, or one of the two daughters was lost (because it left the channel). At the first switch to lactose, a finite induction lag was measured for 90% of the 1,633 cells, while 7% exited the growth channels before inducing, and 3% did not induce during the 4 h in lactose. We note that this *lac* induction lag will include the maturation time of the LacZ-GFP fusion proteins (half time = 5.6 ± 0.4 min [24]). Since we established previously that the fact that rare cells exit the growth channels before inducing does not affect significantly the probability distribution of lags [6], we report here the raw distribution of measured lags.

**Estimation of growth lags.**   When bacteria are exposed to lactose for the first time, they all transiently stop growing. In order to estimate the growth lag $\tau$, i.e., the time until growth restarts after the switch to lactose, from noisy time series of cell length, we devised a Bayesian inference procedure. Let $L(t)$ be the estimated log-sizes across the time points $t$ in the cell cycle from the nutrient switch until the next cell division (after growth has recommenced).

The model we imagine for the log-length assumes no growth until time $\tau$ and perfect linear growth with some constant rate $\lambda$ from time $\tau$ to division. By defining the function,

$$\Lambda^\tau(t) = (t - \tau)\Theta(t - \tau),$$

where $\Theta$ is the Heaviside step function, and by assuming that deviations from our model are only due to Gaussian measurement noise $\epsilon \sim \mathcal{N}(0, \sigma^2)$, we can write

$$L(t) = \lambda\Lambda^\tau(t) + \Lambda_0 + \epsilon,$$

where the constants $\Lambda_0$ and $\lambda$ are the log-size at the growth arrest and the growth rate respectively.

Let a dataset $D = \{L_1, L_2, \ldots, L_T\}$ correspond to the log-sizes of a cell from the time of the nutrient switch ($L_1 = L(t_1)$) until the cell's next division ($L_T = L(t_T)$). The probability of the data $D$ given the parameters ($\Lambda_0, \sigma^2, \lambda, \tau$) is then given by

$$P(D|\Lambda_0, \sigma^2, \lambda, \tau) = \left(\frac{1}{2\pi\sigma^2}\right)^{\frac{T}{2}} \exp\left[-\frac{1}{2\sigma^2}\sum_{j=1}^{T}(L_j - \lambda\Lambda_j^\tau - \Lambda_0)^2\right],$$

where $\Lambda_i^\tau = \Lambda^\tau(t_i)$. We are ultimately interested in inferring the delay $\tau$ only and thus want to marginalize over the other parameters. First, using a uniform prior for $\Lambda_0$ and the Jeffreys

prior for the measurement error $p(\sigma)d\sigma = \frac{d\sigma}{\sigma}$, we can easily marginalize over these two parameters and find

$$P\left(D|\lambda, \tau\right) \propto \left(\mathrm{Var}[L - \lambda\Lambda^\tau]\right)^{\frac{1-T}{2}},$$

where $\mathrm{Var}[L - \lambda\Lambda^\tau]$ corresponds to the average squared deviation between the measured log-sizes $L_i$ and the predicted log-sizes $\lambda\Lambda_i^\tau$.

As far as we can tell, marginalizing over the growth rate $\lambda$ cannot be done analytically, and we therefore used the Laplace method to approximate the corresponding integral over $\lambda$. That is, for a given value of $\tau$, we first determined the optimal value of the growth rate $\lambda^\tau$ which is given by

$$\lambda^\tau = \frac{\mathrm{Cov}[L, \Lambda^\tau]}{\mathrm{Var}[\Lambda^\tau]}. \tag{Eq 1}$$

Expanding the log-likelihood to second order around this optimum in order to approximate the integral over $\lambda$ by a Gaussian integral then leads to the following log-likelihood function as a function of $\tau$ only

$$\log[p(D|\tau)] = \frac{(2-T)}{2}\log[\mathrm{Var}[L - \lambda^\tau\Lambda^\tau]] - \frac{1}{2}\log[\mathrm{Var}[\Lambda^\tau]] + \mathrm{const.}$$

We then compute this log-likelihood numerically for different values of $\tau$ and select $\tau^\star$ to be the one with the highest log-likelihood value. Once we know $\tau^\star$, the other optimal quantities follow directly from the previous calculations and read

$$\lambda^\star = \frac{\mathrm{Cov}[L, \Lambda^{\tau^\star}]}{\mathrm{Var}[\Lambda^{\tau^\star}]}, \quad L_0^\star = \bar{L} - \lambda^\star\bar{\Lambda}_{\tau^\star}, \quad \sigma^{2\star} = \mathrm{Var}\left[L - L_0^\star - \lambda^\star\Lambda_{\tau^\star}\right],$$

where the bar stands for arithmetic mean as usual.

We applied this procedure to the time series of log-length measured after the switch for each cell. Similarly to the induction lags, in order to successfully estimate a lag for as many cells as possible, if a cell divided before inducing, the total length of its two daughters was summed, and cell traces were considered until the total length had tripled since the switch, or one of the two daughters was lost (because it left the channel). At the first switch to lactose, a finite growth lag was measured for 85% of the 1,633 cells, while 9% exited the growth channels before resuming growth, and 6% remain arrested during the 4 h in lactose.

**Estimating LacZ-GFP levels before the switch.** To compare the initial LacZ-GFP levels between cells with short and long lags (Fig 2A), we used the average over the 3 time points after the switch. Since autofluorescence is large relative to LacZ-GFP fluorescence in uninduced cells, it is not possible to detect any correlation between LacZ-GFP levels and the *lac* induction lag (S5 Fig) Another confounding factor is that every experiment has a slightly different offset for fluorescence levels; when pooling observations from different experiments, this difference is larger than the difference of LacZ-GFP levels. Consequently, we compared the distribution of preexisting LacZ-GFP levels in cells with short and long lags after correcting each cell by subtracting the mean in the same experiment (Fig 2A). As a control, we measured the autofluorescence of wild-type cells (gray line in the lower panel) and noticed that the variability of the fluorescence is the same in uninduced cells (blue line), which confirms that it is dominated by autofluorescence (Fig 2A). Comparing the distributions of corrected LacZ-GFP levels revealed that cells with short lags (orange line) have on average 5 LacZ-GFP molecules more than cells with long lags (blue line; Fig 2A).

In order to analyze the sensitivity of the *lac* operon, we would like to know the number of LacZ-GFP molecules in each bacterium at the onset of the switch to lactose. However, the combination of limited camera sensitivity and spontaneous autofluorescence (with fluctuations on the order of ±20 GFP per cell) prevents us from accurately measuring levels lower than 100 molecules. In addition, repeated illumination induces bleaching so that not all GFP molecules are fluorescent (which we previously quantified to correspond to ≈20% per cell cycle) [6]. However, at the second switch, the number of -galactosidases inherited from the ancestors' induction during the first exposure can be computed as follows: The lineage is traced backed in time, and both $L_m$, the maximum LacZ-GFP level during the first exposure, and $n_d$, the number of divisions since this time, are recorded; the expected number of inherited molecules equals $L_m / 2^{n_d}$. Note that this formula assumes no degradation (which is reasonable since the longest delay is 24 h and -galactosidase is stable over this time scale) and even partitioning at every division and that this constitutes a lower bound for the number of -galactosidase molecules in the cell since neither bleaching nor stochastic expression happening during the delay period are accounted for.

**Estimating the LacZ-GFP threshold for induction at low lactose concentration.** In order to measure the LacZ-GFP threshold for induction at lower lactose concentration, we performed experiments where cells with variable levels of LacZ-GFP were switched to minimal media with 0.001% or 0.01% lactose. In order to minimize phototoxicity (expected to be higher due to slower growth and/or longer lags), we reduced the fluorescence excitation intensity 5-fold and the acquisition frequency 2-fold during the period where cells are exposed to lactose. In the experiment with 0.001% lactose, neither induction nor growth was observed in the DIMM. In the 2 subsequent experiments, cells were inoculated at different levels of LacZ-GFP (precultures with IPTG varying between 0 and 20 μM), grown in minimal medium with 0.2% glucose during 6 to 10 h and finally switched to minimal medium with 0.01% lactose.

Data were analyzed using the procedures described above, after correcting for the reduced illumination; note that we did not attempt to correct the corresponding reduction of photobleaching, which may decrease accuracy in this series of experiments. The fluorescence at the switch (used to compute single-cell induction lags) was estimated using the same time window (9 min, hence only 1 data point per cell here) and the number of inherited LacZ-GFP molecules using the equation $L_m / 2^{n_d}$, where $L_m$ is the fluorescence of the ancestor averaged over the first 2 h in glucose and $n_d$ the number of divisions since this time. Finally, we noticed that a large fraction of cells pretreated with IPTG (20 μM) display a visible aggregate of LacZ-GFP in fluorescent images and that all cells of their progeny have long induction lags (even when they have not inherited the aggregate and have measurable LacZ-GFP). Although the underlying mechanism remains unclear, we discarded the corresponding 4 growth channels (out of 62) from subsequent analysis. Overall, the induction lag was successfully measured in 265 out of 287 switching cells.

The lag distribution revealed that short lags span a larger range at 0.01% lactose compared to 0.2% lactose and can last up to 65 min (S8A Fig). Moreover, similar to what we observed at 0.2% lactose (Fig 2C), the fraction of cells with short lags is set by the number of inherited LacZ-GFP molecules, and the threshold for sensitivity remains very low (between 1 and 10 LacZ-GFP molecules; S8B Fig).

## Fluorescence lifetime experiments

**Using FLCS for single-molecule threshold estimation.** In standard FCS, the number of fluorescent particles $N$ in the excitation volume is estimated from the fluctuations of fluorescence intensity along time. This relies on the fact that the autocorrelation of the fluorescence

intensity scales as $1/N$ for short time delays. Noticeably, this method can achieve single-molecule sensitivity. Besides, it is possible to distinguish spectrally overlapping fluorescent species based on their fluorescence lifetime distributions, i.e., the distribution of delay times between excitation and emission, which constitute a unique signature for each fluorophore. This property underlies not only FLIM but also FLCS, which can achieve single-molecule sensitivity in addition.

In FLCS, picosecond time-resolved, time-correlated single photon counting (TCSPC) allows to separate different FCS contributions via their fluorescence lifetime. The principle and mathematical foundations of this lifetime-filtered method can be found elsewhere [29]. Here, the acquisition and analysis procedure for the single-molecule threshold estimation via FLCS performed with SymphoTime64 (PicoQuant) comprised the following steps:

- The lifetime distributions of both GFP and the autofluorescence of cellular compounds were obtained by acquisition of arrival time histograms in point FLCS from induced LacZ-GFP cells (where fluorescence is dominated by GFP) and from cells of the background strain without GFP (where fluorescence derives solely from autofluorescence) (Fig 3A).

- Subsequently, point FLCS was measured in uninduced LacZ-GFP cells. Photon arrival time histograms from these uninduced cells are composed of a superposition of the contributions from autofluorescence components and from GFP, to the extent that GFP molecules are present due to stochastic gene expression.

- Using the previously determined lifetime distributions of the autofluorescence components and GFP, we deconvolve the decaying fluorescence signal by "pattern matching" into mixture of autofluorescence and GFP contributions.

- For each cell, a lifetime-specific autocorrelation curve is calculated from the GFP contribution and fitted to a 3-dimensional purely diffusional model with fitting parameters $N$, the average number of molecules within the detection volume, and $\tau$, the diffusion time across the detection volume. Note that $N$ corresponds to the number of fluorescent particles, which, in the case of LacZ-GFP, are expected to be predominantly tetramers.

- The overall photon count is normalized by $N$ to obtain the apparent photon counts per GFP molecule (CPM). The distribution of CPMs was normalized by its median and the respective frequency distribution fitted to a Gaussian, assuming a random distribution of the error occurring during deconvolution.

- Finally, three different criteria had to be met concurrently to categorize an individual cell as having nonzero LacZ-GFP expression: First, 5 $\sigma$, corresponding to 2.375-fold the median of the CPM distribution, was used as statistical threshold for the lower limit of number of photons truly originating from GFP fluorescence. Second, the physically meaningful range of LacZ-GFP diffusion time was set between 3 ms and 30 ms based on the values obtained from the positive control of induced LacZ-GFP cells. Third, measurements leading to $N > 20$ were discarded under the assumption that more than 20 fluorescent particles would have been detected with FLIM already. Overall, only 5 out of 66 cells analyzed in mother machine experiments with FLCS were classified as dubious for not passing one or several of those criteria (S9 Fig).

**Time-correlated single photon counting setup.** A detailed description of the time-resolved fluorescence microscopy setup can be found elsewhere [30]. In short, FLIM images and lifetime-filtered point FLCS measurements were performed on an Olympus IX73 inverted

microscope stand equipped with a 1.4 NA oil-immersion 100× superapochromat objective (UplanSApo; Olympus) and suitable emission and excitation bandpass filters (Semrock and AHF). A pulsed diode laser (LDH-D-C-485; PicoQuant) was operated at 20 MHz with laser powers set between 25 nW and 100 nW for point FLCS measurements and between 0.25 μW and 1 μW for laser scanning imaging. Emitted photons were detected with a single-photon avalanche diode (SPAD) detector (SPCM CD3516H; Excelitas), and a time-correlated single-photon counting unit (HydraHarp 400; PicoQuant) was used to generate picosecond histograms (16 ps resolution) from the photon arrival times. A galvo scanner (FlimBee, PicoQuant) allowed to record confocal images and to park the confocal volume at an arbitrary $xy$-location within the field of view to perform single-cell point FLCS measurements. Setup control, data acquisition, and raw data analysis were performed with SymPhoTime 64 v.2.4 (PicoQuant).

The entire microscope stand was placed in a temperature incubator (The IceCube, Life Imaging Systems) with the temperature kept constant at 37˚C throughout all experiments. The samples were fixed on a z-piezo stage (Nano-ZL 100, Mad City Labs) on top of a nonmotorized micro-screw $xy$-stage using metal clamps.

**Culture conditions and data acquisition.** For snapshot experiments (Fig 3B), bacteria were grown in M9 minimal media supplemented with 0.2% glucose as described above. Exponentially growing cultures were harvested at OD between 0.05 and 0.3, washed by centrifugation, and concentrated in M9 without sugar. Bacteria were then spread on agarose slabs prepared on glass slides, sealed with a coverslip, and kept on ice until imaging. Prior to point FLCS measurements, images were averaged from 20 to 100 frames of 15 × 15-μm sized scans with 128 × 128 pseudo-pixels and a pixel dwell time of 10 μs. Per image, point FLCS of multiple cells was measured by parking the confocal volume in the center of a cell. Per cell, fluorescence time traces and photon arrival times were recorded for 45 s. Focus stability was monitored from the laser reflection pattern with a separate camera (ZC-F11C4; Ganz).

Mother machine experiments (Fig 3C) were performed as described above (Sections 2.1 and 2.2) with the following modifications that allowed performing the experiment within a day due to the lack of autofocus system; preculture was done during 12 to 14 h, such that cells could be collected in the morning. After inoculation in M9 + 0.2% glucose, cells were allowed 1 to 2 h to recover before starting time-lapse acquisition. Time-lapse acquisition consisted of 40 images every 3 min during the M9 + 0.2% glucose phase and a minimum of 60 images every 3 min after the switch, during the M9 + 0.2% lactose phase. Each image was averaged from 30 frames of 150 × 37.5-μm sized scans with 1200 × 300 pseudo-pixels and a pixel dwell time of 5 μs. Due to the lack of motorized stage and nosepiece, only 1 position could be acquired during each experiment, and the focus was adjusted manually every 5 to 10 min during the entire time-lapse acquisition. Before the switch, point FLCS of up to 2 cells per channel was measured by parking the confocal volume in the center of the cell. Per cell, fluorescence time traces and photon arrival times were recorded for 45 s. Focus stability was monitored from the laser reflection pattern with a separate camera (ZC-F11C4; Ganz).

Using intensity images computed for LacZ-GFP, lag times were recorded manually for each cell measured with FLCS before the switch, as the delay until the cell becomes brighter than uninduced neighboring cells (i.e., until the intensity becomes higher than the background). We note that this criterion differs from the increase by 200 LacZ-GFP molecules used in the epifluorescence experiments, which explains why the cutoff between short and long induction lags here is 35 min instead of 50 min in the former. Only 5 cells exited their growth channel before expressing detectable LacZ-GFP so that their induction lag could not be measured, and 1 cell was discarded for not growing before the switch.

### Diauxie experiments

**Culture conditions.**   All diauxie experiments were performed in 96-well plates (type 655090, Greiner) incubated with 600 rpm shaking. Glycerol stocks were prepared in a single plate for the 8 strains of interest (with 2 replicates for each strain coming from independent clones) and stored at −80˚C. A preculture was inoculated with a pin replicator in M9 + 0.2% glucose and grown overnight to saturation. These cultures were diluted (1,600 to 10,000×, using 2 serial dilutions) to the appropriate growth media, covered with 50 μL mineral oil (to prevent evaporation), and OD was measured approximately every 2 min during 24 to 36 h in a spectrophotometer (Synergy 2, Biotek) with temperature control and shaking ("fast continuous" setting). At least 2 independent replicates were measured per condition.

**Growth curves analysis.**   Wells were inoculated at very low cell concentration so that at least 2 h were recorded with OD below the detection limit. The background absorbance was estimated in each well as follows: the average absorbance was computed in sliding windows of 10 consecutive measurements, and the lowest value of all windows with coefficient of variation below 0.02 was taken as background. For each condition, all growth curves were aligned so that OD = 0.01 at $t$ = 0, where OD = absorbance−background. A few wells showing aberrant records (e.g., discontinuous growth curve) or outlier growth patterns were discarded from further analysis. In addition, strain SC330 displayed very variable growth between replicates (in particular variable lags) and could not be analyzed further since too many growth curves were truncated.

To estimate the population lag for a given condition, we analyze the delay between the corresponding growth curves and the growth curves in the same condition supplemented with 200 μM IPTG. In particular, using simple linear interpolation between the time points at which OD was measured, we determined for each growth curve a continuous function $t = f(\mathrm{OD})$ of the time $t$ for which a particular OD was first reached. Using these, a mean delay and its standard error was computed as a function of OD using all replicates for a given condition (Fig 5). The population lag was then estimated visually as the time at which a plateau occurs in the delay, i.e., the delay value with smallest derivative.

**Diauxie control.**   In order to disentangle the effects of (a) the lactose concentration on *lac* expression in the glucose–lactose mixture and (b) the initial rate of lactose import when glucose gets exhausted, we performed additional experiments where cultures grown on mixtures of glucose and lactose were transferred manually to fresh growth medium with lactose only. Transferring cells between media after washing them required the use of larger volumes; we hence adapted the diauxie protocol to 15 mL Falcon tubes and used only strain ASC662.

Overnight cultures in 2 mL M9 + 0.2% glucose were inoculated from frozen glycerol stocks and grown to saturation with 200 rpm shaking. After 200× dilution to 2 mL fresh media with 0.2% glucose and variable lactose concentrations (no lactose, 0.02%, and 0.2%), bacteria were grown until OD reached 0.08 to 0.2. 1 mL of each culture was then sampled and washed twice in M9 + 0.02% lactose. Each sample was used to start four 150 μL cultures in a 96-well plate, two of which were supplemented with 20% lactose to reach a concentration of 0.2%. All wells were covered with 50 μL mineral oil, and OD was measured approximately every 2 min during 12 h as described above.

### Population lag simulations

Bacterial growth was simulated in discrete time (with 1 min steps) assuming deterministic exponential growth in glucose + lactose and in lactose media, with doubling time of 49 min and 58 min, respectively. Since the effect of nutrient concentration on growth rate was not taken into account, the population size at the start or at the switch to lactose are irrelevant. For

a given empirical distribution of single-cell lags, the distribution was discretized into a 100 sub-populations with different lag values. For each subpopulation, growth was simulated by keeping a constant population until its lag time was reached, followed by simple exponential growth. The population growth curve after the switch was obtained by summing the growth curves of all 100 subpopulations. Finally, the population lag was inferred as the delay compared to the simulation where all cells immediately grow after the switch to lactose, which is measured once all cells have resumed growth.

## Supporting information

**S1 Table. List of experiments used in this study with summary statistics (data from https://git.io/JTS5A).** Experiments that have been discarded from further analysis are grayed out. For experiments acquired before November 2015, data are taken from [6].
(PDF)

**S1 Fig. Schematic of the DIMM microfluidic device.** The DIMM combines a dial-a-wave mixing junction for precise and fast control of the media dispensed to the cells (the left inset shows 3 typical flow regimes: 100/0, 50/50, and 0/100, respectively) and mother machine channels for long-term monitoring of growing bacteria (right inset); adapted from [6]. DIMM, dual input Mother Machine.
(TIF)

**S2 Fig. Physiological state of the cells during the initial growth conditions (data from https://git.io/JTS5A).** For each experiment, the medians and 95% posterior intervals are plotted of both growth rate and cell length at birth. Note that lower illumination during experiments with low lactose concentration leads to higher growth rate.
(TIF)

**S3 Fig. Controls on the statistics of the *lac* operon single-cell lags (data from https://git.io/JTS5A).** (A) Distribution of induction lags for the *lac* operon in naive cells, colored per day. Due to the limited sample size in each experiment, the bin width was increased to 6 min. Note that the bimodality of *lac* induction lag distributions is a robust feature. (B) Distribution of induction lags for the *lac* operon in naive cells, stratified per position in the growth channels. Position is indicated as cell rank, counted from the cell closest to the channel open end in the upper panel and closest to the closed end in the lower panel. Note that cells close to the open end tend to exit the channel before inducing their *lac* operon; hence, the distributions are noisier due to smaller sample size. (C) Distribution of growth lags in naive cells. The bin width is the same as the experimental acquisition frequency (3 min). The corresponding distribution of *lac* induction lags (Fig 1C) is shown for comparison. Note that the distribution of growth lags shows a less marked bimodality, which might result from a combination of less accurate estimation of the growth lag and additional sources of noise being involved in restarting growth once the *lac* operon is expressed.
(TIF)

**S4 Fig. *lac* induction lags for a gradual transition from 0.2% glucose to 0.2% lactose over 40 min (S1 Data, https://git.io/JTS5A).** Comparison of the distribution of induction lags for the *lac* operon in naive cells exposed to a gradual transition from 0.2% glucose to 0.2% lactose over 40 min (2 independent replicates, blue curve) with the distribution of lags under a sudden switch (orange curve, Fig 1C). Note that, since we do not know at what point in the 40 min transition the critical concentrations of glucose/lactose are reached, the lags for each replicate with a gradual transition were offset by a delay that maximized the overlay with the lags under

a sudden switch. The fact that the distribution of lags under the gradual transition is almost identical to the distribution under a sudden switch shows that the stochastic single-cell responses remain equally synchronized under the gradual transition, suggesting that there is a common critical concentration of glucose/lactose across all cells. Due to the limited sample size for the gradual transition, the bin width was increased to 6 min.
(TIF)

**S5 Fig. Induction lag does not correlate with physiological traits at the time of the switch** (S1 Data, https://git.io/JTS5A). Neither cell cycle progression (measured either as the time since birth normalized to the average division time in this condition or as length added since birth as suggested by the "adder" model of cell cycle control) nor fluorescence at the switch (measured in units of LacZ-GFP molecules) nor growth rate correlate with *lac* induction time. Note that fast-switching cells are indicated in orange, and slow-switching cells, in blue. GFP, green fluorescent protein.
(TIF)

**S6 Fig. LacZ-GFP level before exposure to lactose when LacI activity is reduced by low level of IPTG** (S1 Data, https://git.io/JTS5A). Note that this measurements are imprecise due to relatively large fluctuations in autofluorescence (between cells) and in illumination intensity (between replicates). Although this treatment increases the fraction of fast-switching cells, no detectable change of LacZ-GFP levels can be measured which supports that the increase of basal expression is less than 50 molecules. In comparison, bacteria carry 3000 to 6000 LacZ-GFP molecules at full induction (Fig 1B). GFP, green fluorescent protein.
(TIF)

**S7 Fig. *lac* induction lag at the second switch as a function of the estimated number of inherited LacZ-GFP** (S1 Data, https://git.io/JTS5A). Each dot corresponds to a cell with its estimated lag shown along the vertical axis, its estimated number of remaining LacZ-GFP molecules along the horizontal axis, and its color corresponding to the amount of time the cell spend in glucose between the 2 lactose phases. The dotted line corresponds to the 50-min threshold that separates short from long lags. We stratified the cells into 8 groups depending on their estimated numbers of inherited LacZ-GFP molecules remaining at the second switch, and the violin plots show the distributions of lag times of each group, with its horizontal position centered on the average of the group. Note that long lags only reappear for cells with less than 10 inherited molecules of LacZ-GFP. GFP, green fluorescent protein.
(TIF)

**S8 Fig. LacZ-GFP threshold for induction at low lactose concentration (0.01%;** S1 Data, https://git.io/JTS5A). (A) Induction lag as a function of inherited LacZ-GFP (shapes indicate date, and gray dots indicate cells from a lineage with a LacZ-GFP aggregate and are discarded from further analysis). The dotted horizontal line indicates the threshold between short and long lags which is shifted from 50 min (at 0.2% lactose) to 65 min at this lower concentration (0.01%). (B) Fraction of short lags (<65 min) as a function of the estimated number of inherited LacZ-GFP molecules. The dashed horizontal line shows the overall fraction of short lags in naive cells as a reference. GFP, green fluorescent protein.
(TIF)

**S9 Fig. Detection of LacZ-GFP in FLIM experiments (data from** https://git.io/JTSdM). For each cell analyzed with FLCS before the switch in mother machine experiments, 3 criteria must be met concurrently to classify it as GFP positive. Only 5 out of 66 cells where classified as dubious. CPM, counts per molecule; FLIM, fluorescence lifetime imaging; GFP, green

fluorescent protein.
(TIF)

**S10 Fig. Population growth curves during diauxie experiments (data from https://git.io/ JTSd5).** All growth curves used to compute population lags reported in Fig 5B are shown. For each mixture of glucose and lactose (orange lines), the corresponding control with constitutive *lac* operon expression was obtained by supplementing IPTG (blue lines). Each line corresponds to a biological replicate; and delays were computed for OD below the carrying capacity (solid sections). OD, optical density.
(TIF)

**S11 Fig. Lactose concentration during growth in glucose determines the population lag at the diauxic switch (data from https://git.io/JTSd5).** The strain ASC662 was manually transferred from a mixture of glucose (0.005%) and lactose of a given concentration after 4 to 5 h of growth to media with lactose only (OD≈0.1–0.15), at a given concentration. Experiments were performed with 3 different lactose concentrations before the switch and 2 after the switch, for a total of 6 combinations. For each combination, the population lag after the transfer was measured as the delay until OD increased by 0.001 (corresponding to an approximately 10%increase of population size). OD, optical density
(TIF)

**S12 Fig. Many two-component systems in *E. coli* have expression levels so low that no proteins will occur in a substantial number of cells.** Distributions of expression levels of sensor kinases (blue) and response regulator transcription factors (orange) across 28 conditions as measured using quantitative proteomics (data from [18]) for 21 of the 28 two-component systems annotated on ecocyc.org. Each line shows the reverse cumulative distribution of average number of proteins per cell measured across the 28 conditions. The vertical dashed lines correspond to an average of 3 proteins per cell which is the threshold below which more than 5% of the cells are expected not to have any of the corresponding protein (assuming a Poisson distribution for the abundance of low expressed proteins, i.e., $e^{-3} \approx 0.05$).
(TIF)

**S1 Data. Tabular data of single-cell lags measured in mother machine experiments.** This dataset has been extracted from the R working environment available at https://git.io/JTS5A for the convenience of readers willing to use only data on single-cell lags without having to handle their computation and filtering.
(CSV)

## Acknowledgments

We are thankful to all members of the van Nimwegen lab for their input, in particular Chris Field and Guillaume Witz. Image and data analysis were performed at sciCORE, the scientific computing center of the University of Basel.

## Author Contributions

**Conceptualization:** Thomas Julou, Erik van Nimwegen.

**Formal analysis:** Thomas Julou, Erik van Nimwegen.

**Investigation:** Thomas Julou, Ludovit Zweifel, Diana Blank, Erik van Nimwegen.

**Resources:** Ludovit Zweifel, Athos Fiori.

**Supervision:** Thomas Julou, Erik van Nimwegen.

**Writing – original draft:** Thomas Julou, Erik van Nimwegen.

**Writing – review & editing:** Thomas Julou, Ludovit Zweifel, Athos Fiori, Erik van Nimwegen.

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
