## [Editor Report · Decision Letter 0]

24 Jun 2020

Dear Dr Julou, 

Thank you for submitting your manuscript entitled "Subpopulations of sensorless bacteria drive fitness in fluctuating environments" for consideration as a Research Article by PLOS Biology.

Your manuscript has now been evaluated by the PLOS Biology editorial staff, as well as by an academic editor with relevant expertise, and I'm writing to let you know that we would like to send your submission out for external peer review.

Please re-submit your manuscript within two working days, i.e. by Jun 26 2020 11:59PM.

Kind regards,

Roli Roberts

Senior Editor

PLOS Biology

---

## [Decision Letter · Decision Letter 1]

30 Jul 2020

Dear Dr Julou,

Thank you very much for submitting your manuscript "Subpopulations of sensorless bacteria drive fitness in fluctuating environments" for consideration as a Research Article at PLOS Biology. Your manuscript has been evaluated by the PLOS Biology editors, an Academic Editor with relevant expertise, and by several independent reviewers.

As you will see below, the reviewers are all very supportive of the work, although they have nevertheless raised several issues that should be addressed. In light of the reviews, we are pleased to offer you the opportunity to address the comments from the reviewers in a revised version. We anticipate that addressing the issues raised in the report should not take you very long, but please let us know if you expect this will be more than 6 weeks or so. We will then assess your revised manuscript and your response to the reviewers' comments and will likely consult some of the reviewers again.

Please email us (plosbiology@plos.org) if you have any questions or concerns. At this stage, your manuscript remains formally under active consideration at our journal; please notify us by email if you do not intend to submit a revision so that we may end consideration of the manuscript at PLOS Biology.

**IMPORTANT - SUBMITTING YOUR REVISION**

*Resubmission Checklist*

*Published Peer Review*

*PLOS Data Policy*

*Blot and Gel Data Policy*

Sincerely,

Roli Roberts

Senior Editor,

rroberts@plos.org,

PLOS Biology

REVIEWS:

Reviewer's Responses to Questions

PLOS authors have the option to publish the peer review history of their article (what does this mean?). If published, this will include your full peer review and any attached files.

Reviewer #1: No

Reviewer #2: No

Reviewer #3: Yes: Jan-Willem Veening

Reviewer #1: The paper by Julou et al. characterizes the bacterial growth lag during a diauxic shift, at the single-cell level. The authors combine time-lapse microscopy and microfluidics (Dual Input Mother Machine) to visualize the induction of the lac operon in E. coli, when the carbon source is switched from glucose to lactose. They find that the growth lag is very heterogeneous at the single-cell level and exhibits bimodality. Using Fluorescence Lifetime Correlation Spectroscopy (FLCS), they quantify the lac expression level in single cells and show that long lags are due to the absence of beta-galactosidase molecules before the shift. When glucose is switched to lactose, the cells that contain at least one beta-gal molecule can induce their operon while the others have to wait for a stochastic burst of lac expression, thus generating a bimodal distribution of lag times. The authors also show that the growth lag at the population level, which is likely to be an important component of fitness, depends on the fraction of cells with no beta-gal (so-called "sensorless" cells). And this fraction is shown to depend on the environmental conditions before the switch. The authors also present an analysis of proteomics data suggesting that sensorless cells may occur in many other sensory systems, with a potentially important impact on growth lag during environmental perturbations and thus on fitness.

The experiments are well designed and controlled. They are based on cutting edge methods and technologies (DIMM microfluidics setup and image analysis software that the team previously developed), as well as FLCS, which allow a real quantitative analysis. Data analysis is particularly rigorous as well. 

The lac operon has been widely investigated for decades but the growth lag had not been rigorously studied at the single-cell level until now, although it is an important fitness component. And importantly the results are likely to be relevant to other sensory systems in bacteria.

I therefore think this work is an interesting and well executed contribution in the field of stochastic gene expression.

I have a few minor comments:

- I think for figure 2B the colors could be changed (these ones are not very easy to distinguish)

- The legend at the bottom of figure 4D could be improved (for instance what gluc to lac full memory means is not clear and the reader has to find the necessary information in the main text)

-Also I think that fluorescence lifetime imaging is probably classical for physicists but not for biologists and the principle of the technique used here (FLCS) could be explained in more details. I knew the principle of both FLIM and FCS already but I had to struggle a bit to understand what was done here.

Reviewer #2: In this work the authors track single-cell lag of Ecoli growth during the shift from glucose to lactose. This is an extension of Monod's classic diauxie scenario. They find that lag times are bimodally distributed. They then demonstrate that short vs long lag is correlated with pre-existing levels of the LacZ enzyme (which converts lactose to allolactose and thereby allows it to inhibit the LacI repressor). They use multiple strategies to show that the threshold level is of order one molecule.

Upon reading this paper, my first thought was that surely this must have been investigated previously. But indeed, it has not. The authors present many technical improvements (particularly for determination of low LacZ levels, by FLCS, dilution, etc.) in order to develop their results, and this I think makes a very strong case that this paper would be of broad interest: it takes on a classic phenomenon, combines it with ideas that have been informally discussed for a long time, and presents rigorous results.

I have the following major and minor comments

Major:

1. The authors occasionally mention the role of LacY, and that there is a necessary LacY threshold. However, all their measurements are of LacZ-GFP. It is plausible that LacY and LacZ levels are correlated since they are linked in a polycistronic operon. However, especially given the threshold value of 1 LacZ molecule, it would be important to establish the conditional distribution of LacY given a single molecule of LacZ in single cells. Either the claims of the requirement for LacY should be stated as provisional, or direct measurement of LacY is needed. 

2. Could the authors provide for the readers a simple back-of-the-envelope calculation, based on the kinetics of lactose transport by LacY and lactose-to-allolactose conversion by LacZ, the number of pre-existing LacY/LacZ molecules that would be necessary to induce lac operon expression? A related point: the 1-molecule threshold is of course contingent on external lactose concentration (i.e it applies at high lactose concentration, where the transport machinery is saturated). It must be that, at sufficiently low external lactose, the lag time distribution would change because the threshold LacY/LacZ level would change. Could the authors comment on this?

3. What is the possible origin of the 6% of cells that do not grow even with measurable lac operon expression (line 93)? Given the population heterogeneity, 6% is not obviously associated with being "rare".

4. The authors state that the observed dynamics are unrelated to phase of cell cycle. However, since the pre-existing LacZ threshold is of order 1 molecule, there must be cases in which a cell has a short lag and starts to grow, leading to dilution and partitioning of LacZ, in which one of the daughters again receives no LacZ. I understand this is expected to be rare, but how rare? Have any such trajectories been observed in the existing data?

Minor:

Line 147: "surprisingly long lags" -- The sentence is confusing. I assume the authors are referring to the duration of the intervening period, not the growth lag.

Reviewer #3: In the paper by Julou and co-workers, the relevant inducer of the lac operon (lactose) is used to demonstrate that single cell growth lag times towards the switch from glucose to lactose correlate with lac-gene expression activation. By using the non-hydrolysable lactose analog IPTG and overproduction of LacI, they show that there is a causal connection between lac expression and growth lag time upon the nutrient switch. Using automated time-lapse microscopy in microfluidics and fluorescence lifetime correlation spectroscopy they demonstrate that the critical threshold of pre-existing lac expression is on the order of one (LacZ? LacY?) molecule. The authors go on to show that there is not only bimodality in lag times upon switching in the E. coli lab strain but also in non-lab strains. In general, this is a nice study with great figures furthering our knowledge on the lac-operon and demonstrating, to the best of my knowledge, for the first time a direct causal connection between a lag time in cell growth and lac-operon gene expression. Still a few (minor) comments remain.

The authors skip over one key aspect of these findings is that the molecular mechanism behind sensorless cells is the fact that the lac promoter is bound by LacI, and it's the fluctuation in this operator state that determines whether cells stop becoming sensorless. We had a paper on this in 2014: Bhogale et al., NAR, and there are several other papers that might be mentioned. In our 2014 paper we came to the following conclusion for the lac-operon: Fluctuations in the operator state are the rate-limiting fluctuations for the transition to the induced state. It is a combination of two chance events; (i) the repressor unbinding completely from its binding sites on the lac-operon and (ii) the operator sites of the lac-system staying free of repressors for a time sufficient to produce enough importers for positive feedback to kick in and take the cell into the induced state.

I don't like to have to ask the authors to cite one of my own papers, and perhaps there are good reasons not to cite this and other related work by other groups (and I would like to hear them then), but this angle seems pretty relevant to the current study as it is these noisy fluctuations that eventually drive LacZ and LacY production making cells being able to sense and respond (growth) to extracellular lactose.

Ideally (but not necessary), you would like to put the LacZ-GFP in these non-lab E.coli strains and see if again there is correlation in lag time and lacZ-GFP expression. 

L294-305 and Fig. 6. Here the authors attempt to extrapolate analysis from the lac operon to two-component systems present. This last section needs to be expanded and better explained. I get why its here, it's nice to try to expand past the lac operon, but it could be written more clear. Particularly the figure caption, and how the figure is discussed in the text.  Probability of what?

L73, is this a promoter fusion or a translational fusion to the enzyme? Clarify here (later it becomes clear it's a translational fusion). Is this fusion protein still fully functional? Are there different lag times in the parental strain without the lacZ-GFP fusion? One would expect it's probably a bit less functional thereby increasing the proportion of lag times (or the GFP might stabilize the protein and its the other way around).

L80 LacZ-GFP; is this the same construct as described in L73?

L209, CRP acronym first time mentioned.

L238, instead of using 'retarded' perhaps better use delayed?

L252, first use of DIMM abbreviation.

L268 & L269 wherewere

Jan-Willem Veening, University of Lausanne

---

## [Decision Letter · Decision Letter 2]

25 Sep 2020

Dear Dr Julou,

Thank you for submitting your revised Research Article entitled "Subpopulations of sensorless bacteria drive fitness in fluctuating environments" for publication in PLOS Biology. I have now obtained advice from two of the original reviewers and have discussed their comments with the Academic Editor. 

Based on the reviews, we will probably accept this manuscript for publication, assuming that you will modify the manuscript to address the remaining points raised by the reviewers. Please also make sure to address the data and other policy-related requests noted at the end of this email.

IMPORTANT:

a) Please attend to rev #3's remaining request.

b) Please attend to my Data Policy requests further down this email.

We expect to receive your revised manuscript within two weeks. Your revisions should address the specific points made by each reviewer. In addition to the remaining revisions and before we will be able to formally accept your manuscript and consider it "in press", we also need to ensure that your article conforms to our guidelines. A member of our team will be in touch shortly with a set of requests. As we can't proceed until these requirements are met, your swift response will help prevent delays to publication.

- a cover letter that should detail your responses to any editorial requests, if applicable

*Copyediting*

*Published Peer Review History*

*Early Version*

Sincerely,

Roli Roberts

Senior Editor,

rroberts@plos.org,

PLOS Biology

DATA POLICY:

Regardless of the method selected, please ensure that you provide the individual numerical values that underlie the summary data displayed in the following figure panels as they are essential for readers to assess your analysis and to reproduce it: Figs 1BCDE, 2AB, 3ABC, 4BCD, 5AB, 6, S2, S3ABC, S4, S5, S6, S7, S8AB, S9, S10, S11, S12. It is likely that some of these graphs (e.g. growth curves) are plotted directly from the data in your Zenodo and Github depositions; where this is the case, please clearly cite the location of the data in the corresponding legends. NOTE: the numerical data provided should include all replicates AND the way in which the plotted mean and errors were derived (it should not present only the mean/average values).

REVIEWERS' COMMENTS:

Reviewer #2:

The authors have thoughtfully addressed all reviewer comments. They have added fresh experiments which clarify and extend their previous results, all of which support the main thesis of a low molecular threshold for Lac induction. The data provided here will be very useful for further characterization of the Lac system, but also the general principles should be of broad interest.

Reviewer #3:

[identifies himself as Jan-Willem Veening]

The authors did a good job at incorporating my comments. One paper that appeared online during the revision of this work is Moreno-Gamez et al in PNAS (https://pubmed.ncbi.nlm.nih.gov/32669426/), which seems very relevant to the current study and the authors might want to include a few lines on how this work connects (or not) to their work.

---

## [Editor Report · Decision Letter 3]

13 Nov 2020

Dear Dr Julou,

On behalf of my colleagues and the Academic Editor, Nathalie Balaban, I am pleased to inform you that we will be delighted to publish your Research Article in PLOS Biology. 

PRODUCTION PROCESS

Before publication you will see the copyedited word document (within 5 business days) and a PDF proof shortly after that. The copyeditor will be in touch shortly before sending you the copyedited Word document. We will make some revisions at copyediting stage to conform to our general style, and for clarification. When you receive this version you should check and revise it very carefully, including figures, tables, references, and supporting information, because corrections at the next stage (proofs) will be strictly limited to (1) errors in author names or affiliations, (2) errors of scientific fact that would cause misunderstandings to readers, and (3) printer's (introduced) errors. Please return the copyedited file within 2 business days in order to ensure timely delivery of the PDF proof. 

If you are likely to be away when either this document or the proof is sent, please ensure we have contact information of a second person, as we will need you to respond quickly at each point. Given the disruptions resulting from the ongoing COVID-19 pandemic, there may be delays in the production process. We apologise in advance for any inconvenience caused and will do our best to minimize impact as far as possible.

EARLY VERSION

PRESS 

Kind regards,

Alice Musson

Publishing Editor, 

PLOS Biology

on behalf of

Roland Roberts,

Senior Editor

PLOS Biology